# Amniotic LPS-Induced Apoptosis in the Fetal Brain Is Suppressed by Vaginal LPS Preconditioning but Is Promoted by Continuous Ischemic Reperfusion

**DOI:** 10.3390/ijms23031787

**Published:** 2022-02-04

**Authors:** Yupeng Dong, Yoshitaka Kimura, Nobuo Yaegashi

**Affiliations:** 1Advanced Interdisciplinary Biomedical Engineering, Graduate School of Medicine, Tohoku University School of Medicine, Sendai 9818574, Japan; ykimura@med.tohoku.ac.jp; 2Department of Obstetrics & Gynecology, Tohoku University Hospital, Sendai 9818573, Japan; nobuo.yaegashi.c7@tohoku.ac.jp

**Keywords:** chorioamnionitis, bacterial vaginosis, vaginal lipopolysaccharide, fetal brain damage, mouse model

## Abstract

Chorioamnionitis (CAM) is an increasingly common disease affecting pregnant women which derives from bacterial vaginosis. In different clinical cases, it has been shown that CAM can cause multiple risk factors for fetal brain damage, such as infection, and intra-uterine asphyxia. However, the molecular mechanism remains unknown. In this study, we established a novel CAM mouse model by exposing pregnant mice to a combination of three risk factors: vaginal lipopolysaccharides (LPS), amniotic LPS, and ischemic reperfusion. We found amniotic LPS caused Parkinson’s disease-like fetal brain damage, in a dose and time-dependent manner. Moreover, the mechanism of this fetal brain damage is apoptosis induced by amniotic LPS but it was inhibited by being pretreated with a vaginal LPS challenge before amniotic LPS injection. In contrast, amniotic LPS with continuous ischemic reperfusion caused a higher level of apoptotic cell death than amniotic LPS alone. In particular, a potential neuroprotective biomarker phosphorylation (p)-CREB (ser133) appeared in only vaginal LPS preconditioned before amniotic LPS, whereas ischemic reperfusion triggered IKK phosphorylation after amniotic LPS. Despite the need for many future investigations, this study also discussed a developed understanding of the molecular mechanism of how these phenotypes occurred.

## 1. Introduction

The prevalence of Bacterial vaginosis (BV) in the general population is high globally, ranging from 23% to 29% across regions (Europe and Central Asia, 23%; East Asia and Pacific, 24%; Latin America and Caribbean, 24%; Middle East and North Africa, 25%; sub-Saharan Africa, 25%; North America, 27%; South Asia, 29%) [1,2,3,4]. Chorioamnionitis (CAM), an increasingly common intrauterine infection often derived from ongoing BV, has been shown to have a positive correlation with cerebral palsy [5,6]. Although transplantation of fetal brain cells has been used as a potential therapy for Parkinson disease, fetal brain damage or cerebral palsy, may result from different risk factors including gestational age of <32 weeks, low birth weight, asphyxia, infection, and inflammation induced by lipopolysaccharide (LPS) [7,8,9,10,11,12]. In the case of infection (viral or bacterial), there are cytokines produced in response to bacterial or viral products. These products include lipopolysaccharide and others. However, the onset of fetal brain damage varies depending on the condition. The causes of cerebral palsy in term and preterm babies are different. Compared with other models, such as the preterm brain damage model [13,14], the infectious preterm chorioamnionitis model [15], and the LPS induced maternal systematic to fetal inflammation model [16,17,18,19,20,21,22], our mouse model was only designed to mimic increasing inflammation at term. It is critical to model asymptomatic BV-like vaginal inflammation to intrauterine inflammation following uterine contractions-like ischemic reperfusion. Reviews have shown that it is hard to attain consensus by comparing individual studies, as during brain development, infection and inflammation or ischemic reperfusion, have a variable pathological course, resulting in a range of phenotypes [11,23]. For example, evidence from animal models has shown that the bacterial toxin LPS could induce different types of fetal brain injury [10,11,12,17,21,23,24,25]. On the other hand, dosage, or administration of LPS has also been shown to have a neuroprotective effect against acute inflammation and asphyxia in the fetal brain [26,27,28]. As conflicting opinions remain, further investigation of the molecular mechanism is required to clarify the pathological processes of fetal brain damage induced by multiple factors. 

LPS activates two major inflammatory response pathways: nuclear factor-kappa B (NF-κB) and mitogen-activated protein kinase (MAPK) signaling [29,30]. The NF-kB pathway is commonly inhibited by IkB during pregnancy. IKKalpha and beta are the catalytic subunits of the kinase of IkB. Phosphorylation of these proteins may result the NF-kB activity induced apoptosis, inflammation and preterm birth. The MAPK pathway has three subfamilies, ERK, P38 and JNK. Phosphorylated (p)-ERK is assumed to play a role in cell proliferation and survival, while p-P38 or p-JNK1/2 is involved in stress-response signaling [31,32]. Phosphorylation of CREB is considered as a survival signal and has a potential neuroprotective role through its transcriptional role [33,34,35,36,37,38,39,40,41,42]. This study used a continuous mouse model to comparably investigate the changes of molecular expression level in a classic pathology process of CAM. The goal was to understand whether and how vaginal LPS regulates the vulnerability of the fetus to brain damage secondary to those risk factors that might be present in CAM, e.g., amniotic infection and ischemic reperfusion.

## 2. Results

### 2.1. Amniotic LPS-Induced Apoptotic Cell Death Is Promoted by Continuous Ischemic Reperfusion

As described in materials and methods, a mouse model including three risk factors of fetal brain damage was used for this study (Figure 1A). We investigated the effect of amniotic LPS varying the dose. Ten microliters of 1 mg/mL LPS per fetus induced a significant acute response, as evident in increased Parkinson-disease 7 also known as DJ-1 (Figure 1B). FECG was able to detect fetal heart rate, significantly reduced by ischemic reperfusion (Figure 1C), and continuous extensions of the R–R interval more than 1 min without recovery (arrhythmia) 3 h after LPS injection in amniotic fluid (AF3h in Figure 1D). 

However, it was too late to help the fetus because amniotic LPS-induced fetal death had been initiated in the AF3h (Figure 2A). Even in surviving AF3h fetuses, using the apoptotic marker protein, i.e., cleaved poly (ADP-ribose) polymerase (Parp), a significant increase of cleaved Parp in AF3h was confirmed (Figure 2B,D). Fetal brains also had widespread apoptotic cell death (Figure 2B,C). Moreover, a higher level of cleaved Parp was present in AF3h following continuous ischemic reperfusion at the endpoint of 3 h (AF3hIR in Figure 2D). In contrast, AF3h did not significantly increase autophagy protein LC3A/B compared with other conditions (Figure 2E). 

### 2.2. Amniotic LPS-Induced Apoptotic Cell Death Is Inhibited by Vaginal LPS Preconditioning

As for the above Parkinson associated protein DJ-1 data (Figure 1B), to investigate the fetal brains’ condition we also used alpha-synuclein, a protein also linked to Parkinson’s disease [43,44,45,46,47]. In contrast to VIR and VAFIR, VAF showed a significantly inhibited alpha-synuclein (SNCA) when compared with AF (Figure 2E). 

Next, to increase the sample quality, selected fetal brains derived from four conditions: N, V, AF, and VAF were used for further investigation. First, a time-course analysis of cell death was performed (Figure 3A). At 0.5 h, there were no detectable morphological changes between the experimental conditions (Figure 3B). Apoptotic-like positive cells became detectable (brown area in Figure 3C and blue triangles in Figure 3D) after 1.5 h from amniotic LPS injection, and there was significantly more apoptotic cell death in AF compared with the N and VAF groups.

### 2.3. Vaginal LPS Preconditioning Rescues the Amniotic LPS-Induced Apoptosis through Cell-Signaling Transduction Pathway and Transcriptional Factors

To understand the molecular mechanism of the above phenotypes, cell-signaling pathways and transcriptional factors were investigated (Figure 4). In our mouse model, amniotic LPS administration increased the phosphorylation of JNK (area 3, 4, 5, and 6 in Figure 4A; the last graph in Figure 4B) and reduced the phosphorylation of ERK1/2 (Figure 4B) at AF0.5h. Vaginal LPS increased p-P38 and partly reduced amniotic LPS-induced p-JNK (Figure 4A,B), specifically p-JNK2 (Figure 4C). Also, there was a p-JNK/p-ATF cooperation activated response to the conditions (Figure 4D,E). 

Next, we further investigated the activation of the downstream cell-signaling transcription factors, including ATF, P53, HIF1-α, stat3, and cAMP response element-binding protein (CREB). CREB is between the HIF1-α and P53 (Figure 5A) these two transcription factors have a transcription switch under ischemic reperfusion (21). CREB phosphorylation at Ser133 was increased in response to continuous stresses with vaginal LPS preconditioning (Figure 5B,C). These results supported our hypothesis that vaginal LPS preconditioning protected the fetal brain from amniotic LPS-induced apoptosis.

In contrast, severe ischemic reperfusion induced the phosphorylation of IKK (Figure 5A,B). 

### 2.4. Vaginal LPS Preconditioning Results in an Earlier Inflammatory Response to Amniotic LPS including IL-6 Releasing in Fetal Brain

In the fetal brain, expression of inflammatory cytokines, such as TNF-α, IL-1, and IL-6, was higher in the VAF group compared with the AF group (Figure 6A). Moreover, IL-6 increased in the mother’s plasma in a time-dependent manner and was significantly early appeared under VAF than AF condition (Figure 6B). Moreover, the IL-6 protein was released at higher levels in the amniotic fluid and fetal plasma than in the mother’s plasma (Figure 6C). 

## 3. Discussion

This study was based on the experimental LPS mouse model. Only LPS induced inflammation, not infection by bacteria or fungi. However, this study design also removed the noise caused by infection.

The fetal condition became critical after amniotic LPS administration, such that Parkinson-like fetal brain injury (DJ-1 in Figure 1B and SNCA in Figure 2E) and fetal death occurred (Figure 2A), in both concentration- and time-dependent manners. Notably, The DJ-1 data indicated that there is a link between amniotic LPS maybe through its induced systemic inflammation and Parkinson-like fetal brain injury. A promising treatment strategy is the FECG, which also allows real-time monitoring of the extension of the R–R interval (Figure 2D). Although little is known about this novel technology [48,49,50,51], it is undergoing clinical trials in Japan and is expected to undergo international distribution (www.fetalecg.med.tohoku.ac.jp. Accessed date 6 March 2021). The death of the fetuses and abnormal fetal electrocardiographs appeared in the AF3h group, but not in the VAF group (Figure 1D and Figure 2A). We also proved that the mechanism of amniotic LPS induced Parkinson disease-like fetal brain damage was apoptosis [52] through TUNEL staining and apoptosis biomarker cleaved-Parp analysis. Moreover, our data also implied that the condition was worse in the AF group than in the VAF group (Figure 2). VAF reduced this kind of apoptosis (Figure 3C,D), but Ischemic reperfusion enhanced the amniotic LPS-induced apoptosis in the fetal brain (Figure 2B,C).

In this study, vaginal LPS preconditioning of the fetal brain activated the P38 pathway, whereas JNK was activated by amniotic LPS (Figure 4A,B). JNK2 has been considered as a negative regulator of cell proliferation signaling [53] and mainly a response to LPS in microglia [54]. Our data showed that VAF more reduced the JNK2 activity than AF (Figure 4A–C). VAF also significantly rescued the amniotic LPS inhibited p-ERK in the whole brain extract (Figure 4B). This maybe explains how vaginal LPS preconditioning prevents fetal brain apoptosis induced by amniotic LPS in the early period. Consistent with their relationship in the correlational distance data (blue line in Figure 4E), p-CREB (Ser133) is a potential neuroprotective biomarker [35,36,37,38], whereas p-ATF2 (Thr71) was considered as a fetal brain hemorrhage biomarker under the VIR condition [21]. We first reported that p-ATF1/p-CREB (Ser133) presented in the VAF group (Figure 5), whereas p-JNK/p-ATF2 (Thr71) and p-ATF1/p-CREB (Ser133) were activated in AF3h (Figure 4D). These data implied that, although the rescue system (the activation of p-ATF1/p-CREB pair) was working at both VAF and AF3h conditions, AF3h was a worse condition (the activation of p-JNK/p-ATF2 pair) than VAF. 

Proinflammatory cytokines IL-1, IL-6, and TNF-α have both positive and negative roles of brain injury in the fetus (17, 58, 59). At the endpoint, 30 min after amniotic fluid LPS injection, the expressions of these cytokines in fetal brains were higher in VAF than in AF (Figure 6A). Supporting evidence was confirmed by both IL-6 mRNA, and protein levels analysis. They were higher in VAF as an early immune reaction (until 3 h in Figure 6B) compared with the AF group. On the other hand, our data also showed a higher activation of ERK and STAT (Appendix A) in VAF than in AF; which regulates the expression of IL-6 that implied rescue signaling transductions, so that, AF and VAF all have no detectable apoptosis at the endpoint 30 min after amniotic LPS injection. In addition, Figure 6C data suggests that a higher concentration of IL-6 was in the plasma from the fetus than from the mother, explaining why for this phenotype all fetuses were dead in 24 h without the mother mouse dying (Figure 2A).

However, to understand the relevance of these results to clinical practice needs further investigation. In this study, we suggested that the combined utilization of FECG and fetal brain damage imaging by using p-CREB/p-IKK biomarkers, it may be a novel approach for supporting clinical therapy.

## 4. Materials and Methods

All animal experiments were conducted in accordance with the guidelines for animal experimentation of the Tohoku University, Sendai, Japan. The Tohoku University Committee for the Safety Management of Animals approved all experimental protocols.

### 4.1. Mouse Model

C57BL/6N mice (CLEA Japan Inc., Tokyo, Japan) were bred overnight (from PM 5:00 to AM 8:00), marking gestational day 0 (GD0).

Pregnant mice were randomly placed under eight conditions (Figure 1) as listed below:Native(N): no treatmentVaginal (V) LPSAmniotic fluid (AF) LPSIschemic reperfusion (IR)Vaginal LPS plus amniotic fluid LPS (VAF)Vaginal LPS plus ischemic reperfusion (VIR)Amniotic fluid LPS plus ischemic reperfusion (AFIR)Vaginal LPS plus amniotic fluid LPS plus ischemic reperfusion (VAFIR)

A mouse model of increasing intrauterine infection was developed using three steps (Figure 1A):Step 1. On both GD14 and GD16, vaginal LPS was administered (10 μL of 1 mg/mL LPS, 95% confidence interval was 0.44 ± 0.14 mg/kg per pregnant mouse, *n* = 12 pregnant mice) using a Pipetman (Gilson, Inc., Middleton, WI, USA).Step 2. The temperature and humidity were controlled at 36 °C ± 1 °C and 60% ± 5%, respectively, during the cesarean section (Appendix A). All surgical tools were autoclaved at 120 °C for 20 min. GD18 pregnant mice were anesthetized with subcutaneous 50 mg/kg ketamine (Ketalar 500 mg^®^; Daiichi-Sankyo, Tokyo, Japan) and 5 mg/kg xylazine (Rompun™ 2% solution for injection; Bayer AG, Leverkusen, Germany) and maintained on inhalational isoflurane (Forane^®^ 0.5%, 260 mL/min; AbbVie Inc., Chicago, IL, USA). After scrubbing in and putting on surgical gloves, the fetal electrocardiography (FECG) sensor was cleaned using 100% ethanol for 10 min before surgery (Appendix A). Povidone iodine (7.5%) was used as an antiseptic on the skin of the mice before dissection and for the sterilization of surgical instruments. After the abdominal operation, 10 μL of LPS solution (1 mg/mL, with 95% confidence interval: 11.43 ± 4.03 mg/kg per fetus, *n* = 12 fetuses derived from 6 dams) was injected into each hole of amniotic fluid using sterile needles (30G) and syringes (BD, Franklin Lakes, NJ, USA), and the wounds were closed within 5 min. Fetal brains were then collected after 3 h (AF3h; *n* = 6 pregnant mice), or 24 h (AF24h; *n* = 6 pregnant mice), from amniotic fluid LPS injection for a time-course pathology analysis. If necessary, the condition of the fetuses was continuously monitored using a FECG sensor until the endpoint of 3 h after LPS injection. After surgery, pregnant mice were returned to their cages for the indicated time. For surgery 24-h samples, a 12-h on/12-h off light cycle was used, and temperature and humidity were maintained at room temperature.Step 3. Thirty minutes (except 3h in AF3hIR) after the amniotic LPS injection, ischemic reperfusion was performed by clipping the uterine artery for 5 min with a release for 5 min and repeating this for three cycles. Specifically, repeated twice normal IR (3 cycles), six cycles of clipping and release were performed as a worse ischemic reperfusion condition (*n* = 3 pregnant mice). Electromyography (EMG) was used to measure uterine muscle contractions. One EMG cycle approximated to 1 min, including a 20-s contraction and 40-s relaxation. As described in previous reports [55,56], FECG is sufficiently sensitive to monitor asphyxia-like ischemic reperfusion (Figure 1C). However, in the early period of 30 min, only amniotic LPS either without or with vaginal LPS preconditioning was hard to detect based on raw FECG data.

### 4.2. Quantitative and Qualitative Analyses of Fetal Mouse Brains

The fetuses showed a great variation in weight. There was a significant weight difference between fetuses derived from the same dam. Moreover, the number of fetuses in each dam varied from 1 to 13. Based on these observations, a qualitative selection of fetal brains was considered necessary for comparing the results of molecular analyses between groups. To prepare the AF and VAF conditions with the N and V conditions, 433 fetuses (353 survived, 80 died) derived from 70 pregnant mice, were selected according to three filters:Number of fetuses in a dam: 4–9Fetal body weight: 850 ± 150 mgFetal brain weight: 50 ± 5 mg

For all conditions involving ischemic reperfusion (IR, VIR, AFIR, and VAFIR), the uterine arteries feeding at least three fetuses from one uterine horn were clipped. A total of 136 surviving fetuses derived from 32 pregnant mice were collected. Some fetal brains were derived from the same dams used in previous publications [55,56].

### 4.3. Western Blotting

For western blotting, fetal brains were carefully collected (Appendix A), flash-frozen in liquid nitrogen, and stored at −80 °C. As detailed in a previous report [55], total extracts of each fetal brain were admixed with 1% NP40 buffer (Cell Signaling Technology, Inc., Danvers, MA, USA) for western blotting. The antibodies (1:1000) used for western blotting were purchased from Cell Signaling Technology, Inc. Western blotting data were analyzed using ImageJ (National Institutes of Health, Bethesda, MD, USA).

### 4.4. Quantitative Real-Time Polymerase Chain Reaction

For quantitative real-time polymerase chain reaction (qRT-PCR), fetal brains were carefully collected, flash-frozen in liquid nitrogen, and stored at −80 °C. Total RNA was extracted as described in a previous report [56]. In particular, the first-strand cDNA was synthesized using a SuperScript^®^ III First-Strand Synthesis SuperMixKit (18080400; Invitrogen, Carlsbad, CA, USA). Using 1 µL of cDNA (1:100 diluted in autoclaved ultrapure distilled water), we performed qRT-PCR comprising 40 cycles at 60 °C. The primers used are listed in Appendix A. The housekeeping gene Hprt was used to calculate for each fetal brain. ∆Ct = Ct genes − Ct Hprt. ∆∆Ct (A/B) = ∆Ct condition A − ∆Ct condition B. Folds (A/B) = POWER (2, ∆∆Ct).

### 4.5. Enzyme-Linked Immunosorbent Assay

Maternal or fetal blood samples were collected into precooled tubes containing EDTA or heparin centrifuged at 2200 g for 15 min at 4 °C. The supernatants (plasma) were stored at −80 °C until analysis. Amniotic fluid or fetal blood samples were collected from multiple fetuses derived from one dam (a dam was considered as *n* = 1). A Mouse Inflammatory Cytokines Multi-Analyte ELISArray Kit (MEM-004A; SABiosciences, Germantown, MD, USA) was used to detect 12 types of proinflammatory cytokines in the plasma or amniotic fluid sample. The cytokines in the array were interleukin (IL)-1A, IL-1B, IL-2, IL-4, IL-6, IL-10, IL-12, IL-17A, interferon-γ, tumor necrosis factor-α (TNF-α), granulocyte colony-stimulating factor (G-CSF), and granulocyte–macrophage colony-stimulating factor (GM-CSF). Only positivity or negativity, according to the optical density at 600 nm, could be detected. For further analysis, the concentration of IL-6 (ab46100; Abcam, Cambridge, UK) was measured according to the manufacturer’s protocol.

### 4.6. Immunohistochemical Scoring

The whole fetal brain, including bone, was fixed in 15 mL of 4% paraformaldehyde for 3 days. After dehydration in 80%, 95%, and 100% ethanol, paraffin-embedded tissue was prepared. Fetuses were grouped based on the body and brain weights and compared. For this evaluation, the whole fetal brain was separated into six areas (Fields 1–6). The hippocampus for Field 3 or lateral ventricle for Field 6 (white matter) were used as landmarks. Field 4 was at the top of the white matter, and Field 5 was at the bottom of the white matter. Fields 1 and 2 contained the cerebral neocortex.

A histological score of each field was developed to assess the degree of immunohistochemical staining. The following classification was used: 0, negative; 1, unclear or unknown; 2, positive; 3, strong or wide staining; and 4, strong plus wide staining. For each localized field, the relative grade for each group was calculated as follows:

Grades = (average number of left and right side) × (four fetuses derived from at least three pregnant mice).

### 4.7. Imaging Experiments

For total imaging of the fetal brain, hematoxylin and eosin staining was used. A mouse on mouse kit was used (BMK-2202; Vector Laboratories, Inc., Burlingame, CA, USA) with anti-mouse primary antibodies in each section to evaluate the presence of antibody-positive cells (×1.25 to ×100 objective lenses). Antibodies were diluted before use (Appendix A). For detecting immunofluorescence, secondary antibodies conjugated with AlexaFluor^®^ 488 or Cyanine 3 dye (Life Technologies, Carlsbad, CA, USA) and a 780 confocal microscope (Carl Zeiss AG, Oberkochen, Germany) were used. An FSX100 microscope (Olympus Corp., Tokyo, Japan) was used to produce 5 × 5 multiple pictures using ×20 objective lenses. Continuous pictures were individually acquired using ×100 objective lens and stitched together to create a map of the fetal brain according to the manufacturer’s protocol. For immunochemical experiments, sections of 3 μm were stained according to the primary antibody datasheet. For counterstaining, hematoxylin or methyl green was used.

### 4.8. Terminal Deoxynucleotidyl Transferase dUTP Nick-End Labeling Assay

Based on the in situ cell death detection (POD) kit manual (Instructions for use-version 14, 11684817910; Roche Applied Science, Penzberg, Germany), at least three different fetuses derived from three pregnant mice in each group were used to quantify cell death. Before antigen–antibody reaction studies, slides were incubated in a permeabilization buffer (0.1% Triton X-100 in 0.1% sodium citrate) for 8 min. After 3,3′-diaminobenzidine tetrahydrochloride (D006; Dojindo Laboratories Co., Ltd., Kumamoto, Japan) staining, death-positive cells (brown) were confirmed by morphologic analysis using microscopy (×100) to classify cells as being proliferative, metabolic, apoptotic, or necrotic. Methyl green was used as the counterstain.

### 4.9. Transmission Electron Microscopy

Pregnant mice were immobilized by perfusion fixation (2.5% glutaraldehyde and 4% paraformaldehyde in 0.1 M cacodylate buffer) after anesthesia. Fetal brains within the intact cranium were fixed using 2 mL of 25% glutaraldehyde (EM grade, G004; TAAB Laboratory Equipment Ltd., Reading, UK) with 10 mL of 0.2 mol/l cacodylate buffer (036-18175; Wako Pure Chemical Industries, Ltd., Osaka, Japan) in 8 mL distilled water PB buffer (pH 7.4) at 4 °C for 5 days (Appendix A). The brains were sliced into areas and washed using 0.1 mol/L cacodylate buffer for 15 min at least three times. The slices were incubated in 1% osmium tetroxide for 90 min on ice followed by incubation in 1% uranyl acetate solution for 50 min at room temperature. The slices were carefully dehydrated using 50%, 60%, 70%, 80%, 90%, and 95% ethanol for 10 min and 100% ethanol for 20 min. Pretreatment with propylene oxide (PO) for 10 min was performed twice. Tissues were embedded in 1:1 PO resin (Embed-812, DDSA, NMA, 2% DMP-30) for 60 min, followed by 1:2 PO resin overnight. Cut sections of 70–80 nm thickness were arranged on Formvar/carbon-coated nickel grids. After each section was dried, it was evaluated (magnification: ×700–10,000) using a transmission electron microscope (H-7600; Hitachi Ltd., Tokyo, Japan).

### 4.10. Statistical Analysis

The parameters derived were expressed as mean ± standard error of the mean. The statistical method [15] used was the Student’s *t*-test (two-tailed for two independent groups). For multiple group comparisons, statistical analysis was performed using the Kruskal–Wallis one-way analysis of variance. A probability *p* value of <0.05 was considered significant. Statistical analysis was performed using GraphPad Prism 5 (GraphPad Software, Inc., La Jolla, CA, USA).

## 5. Conclusions

The preactivated transcriptional factors such as p-CREB by vaginal LPS were triggered to respond to the secondary risk factors such as amniotic and/or ischemic reperfusion. Moreover, MAPK-ATF/CREB were potential targets to rescue fetal Parkinson disease-like brain injury induced by LPS inflammation.

## Figures and Tables

**Figure 1 ijms-23-01787-f001:**
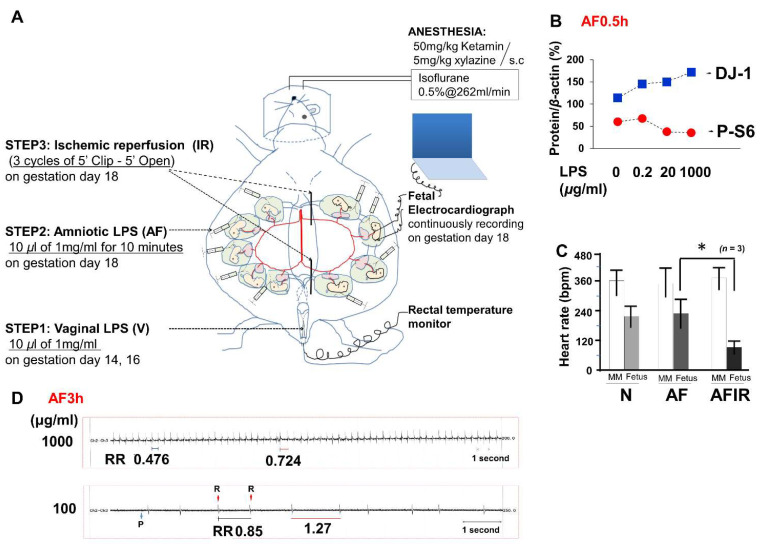
Fetal electrocardiography detects ischemic reperfusion. (**A**) A novel mouse model to expose fetuses to vaginal lipopolysaccharide (LPS) preconditioning, amniotic LPS, and ischemic reperfusion from gestational days (GDs) 14–18. On GD18, fetal electrocardiography (FECG) was used as a real-time monitor. (**B**) Western blot assays of amniotic fluid (AF) LPS 0.5 h fetal brain tissue lysate (1% NP40) to detect Parkinson-disease 7 (PARK7, also known as DJ)-1 under indicated LPS concentration. (**C**) Heart rate detected by FECG of indicated conditions including ischemic reperfusion after exposure to lethal amniotic LPS (AFIR). * *p* < 0.05. (**D**) FECG of fetuses (AF condition) exposed to amniotic LPS at both 1 mg/mL and 0.1 mg/mL concentrations.

**Figure 2 ijms-23-01787-f002:**
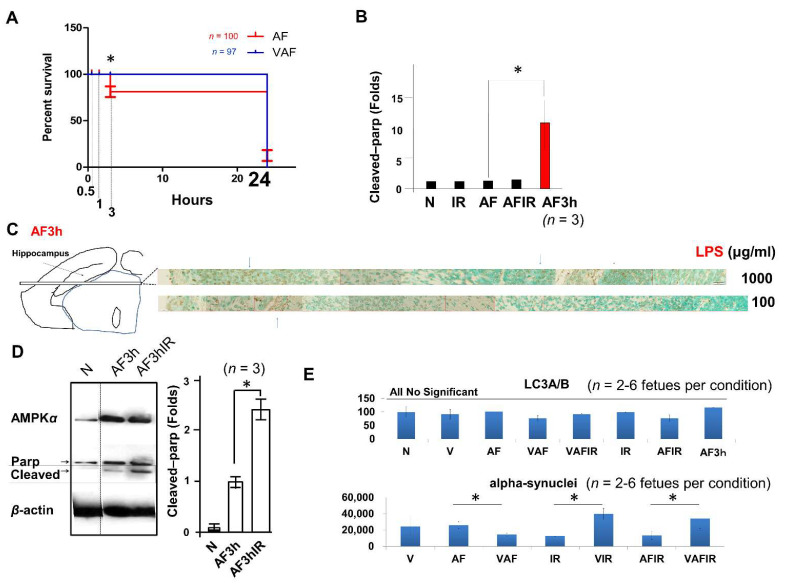
Amniotic LPS causes fetal death. (**A**) Dead fetuses were counted in the AF (red) and VAF (blue) groups from the endpoint 3 h. Almost all fetuses died by liquefactive necrosis of brains after 24-h exposure to amniotic LPS, even after vaginal LPS preconditioning. (**B**) An increased cleaved Parp protein (10 folds) supports the apoptosis-like cell death in the AF. (**C**) Significant cell death occurred in indicated concentrations (AF). (**D**) Protein level of cleaved Parp in the indicated conditions (**E**) Protein level of LC3A/B and alpha-synuclein (SNCA) in the fetal brain at different condition by western blot. * *p* < 0.05.

**Figure 3 ijms-23-01787-f003:**
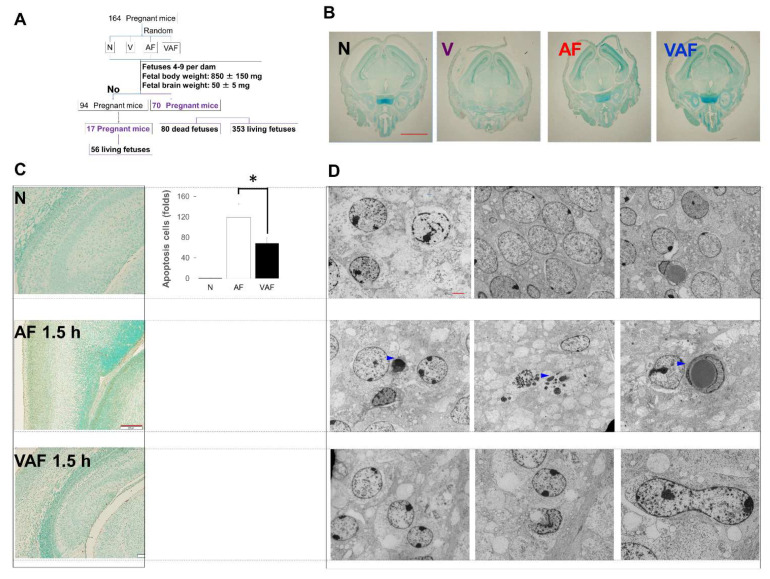
Time-course investigation of cell death in the fetal brain. (**A**) Quantitative and qualitative analyzes of fetal mouse brains, details in materials and methods. (**B**) No detectable cell death in the whole fetal brains of N, V, AF 0.5 h and VAF 0.5 h (×1.25 microscopy). Scale bar means 2 mm. (**C**) 1.5 h of AF and VAF cell death measure (brown doses are positive apoptosis cells). Scale bar means 200 μm. * *p* < 0.05. (**D**) Transmission electron microscopy images of cell death in the Native, Apoptosis [blue triangles in AF (1.5 h), Scale bar means 2 μm.

**Figure 4 ijms-23-01787-f004:**
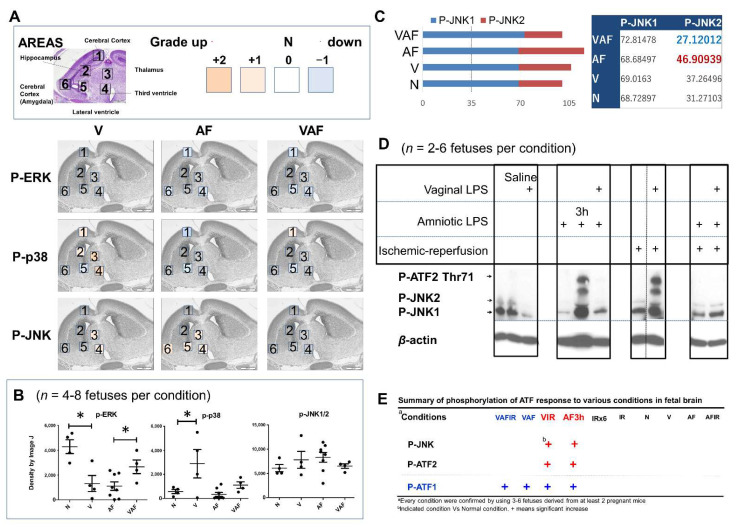
The phosphorylation of mitogen-activated protein kinase pathways and transcription factors. (**A**) Three fetuses derived from three pregnant mice per group were used for immunochemical staining. Each fetal brain was separated into six fields (1–6). The grades (0–4) of phosphorylated protein in the paraffin-embedded sections areas, which we indicated using different colors. (**B**) Western blot data of phosphorylated ERK1/2, P38, and JNK1/2. * *p* < 0.05. (**C**) The average percentage of phosphorylation of JNK1 and JNK2, normal condition (N) is 100%. (**D**) Phosphorylated ATF2 and JNK in the whole fetal brain under indicated conditions. (**E**) Summary of western blot data: strong phosphorylation (++) of both JNK and ATF conditions were indicated.

**Figure 5 ijms-23-01787-f005:**
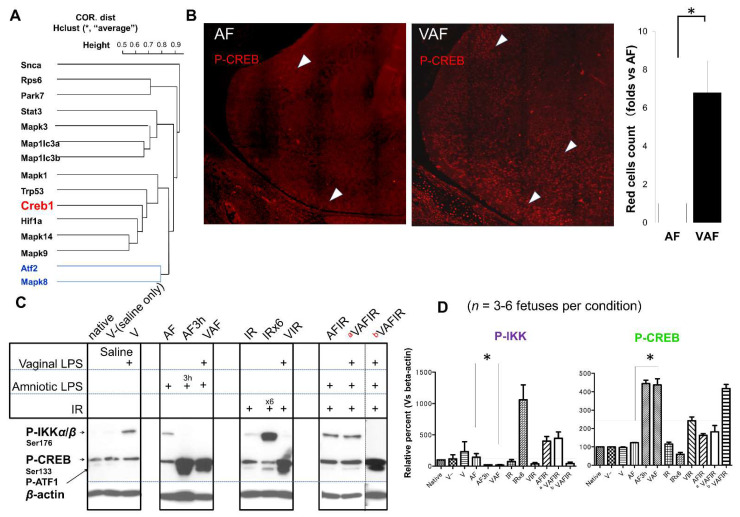
The phosphorylation of CREB was activated in VAF group but not in AF group. (**A**) hclust average of correlation distance of indicated proteins. (**B**) Immunochemistry images of phosphorylated CREB (ser133) in fetal brains derived from the amniotic LPS and VAF groups. * means *p* < 0.05. (**C**,**D**) Phosphorylated CREB and IKK were measured by western blotting. The bands under P-CREB are phosphorylation of ATF1. β-actin was used as a loading control. Samples from the VAFIR condition showed two distinct molecular phenotypes described as ^a^ VAFIR and ^b^ VAFIR respectively. * *p* < 0.05.

**Figure 6 ijms-23-01787-f006:**
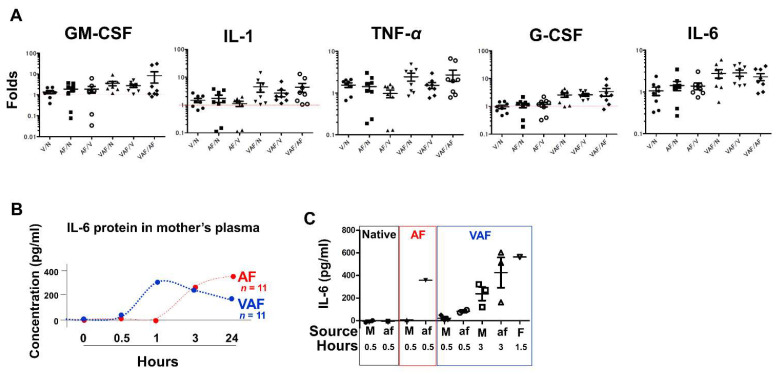
Cytokine expression measured by ELISA. (**A**) The relative folds of cytokine gene expression by comparing with the indicated conditions, V: vaginal; N: normal/native; AF: amniotic LPS; VAF: vaginal LPS plus amniotic LPS. Hprt is a housekeeping gene used as a control. (**B**) IL-6 protein in the maternal plasma (M) was measured at each endpoint time (0–24 h) after amniotic LPS exposure. Either saline injection into the amniotic fluid, or the vaginal LPS group, were used as control samples to compare with the AF (red) or VAF (blue) groups, respectively (for detailed information refer to raw data). (**C**) IL-6 concentration in M plasma and amniotic fluid (af) at the indicated time. Fetal plasma (F) represents a mixture derived from 12 fetuses from 3 dams.

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
