# Peer review of "Amniotic LPS-Induced Apoptosis in the Fetal Brain Is Suppressed by Vaginal LPS Preconditioning but Is Promoted by Continuous Ischemic Reperfusion"

_ijms, 2022, doi:10.3390/ijms23031787_

Round 1

Reviewer 1 Report

In this manuscript which the title is “Amniotic LPS-induced Apoptosis in the fetal brain supressed by vaginal LPS preconditionning but is promoted by continuous ischemic reperfusion”, Dong et al. aimed to explore the complex pathophysiology of chorioamnionitis that led on various consequences on the fetus brain.

They used a pregnant mouse model based on three experimental steps including vaginal LPS (10 ng) on gestational days (GD) 14 and 16, LPS injection in the amniotic fluid on GD 18 and repeated transient interruptions of the uterine artery flow for 30 min just after LPS on GD 18 mimicking ischemic-reperfusion cycles during uterine contractions. Eight experimental groups resulted for this experimental schedule.

The LPS injections in the amniotic fluid revealed to be widely fatal for fetuses in this model in a short delay (3 hours) and with 100% of mortality at 24 hours. Deep bradycardia of fetuses were suggested 3 hours after LPS injection in the amniotic fluid.

Apoptotic cell death was detected in fetal brains from 1,5 hour after LPS injection and was more intense in association with the ischemic-reperfusion insult. 

The vaginal exposure to LPS prior to LPS injection in amniotic fluid increased the fetus brain expression of p-ERK, p-38 and decreased p-JNK1-2 in comparison with fetuses who only received LPS in amniotic fluid. Furthermore, vaginal exposure to LPS prior to LPS injection in the amniotic fluid promoted p-CREB expression. These data were interpreted by authors as a preconditioning effect of vaginal LPS. Surprisingly, vaginal exposure to LPS alone induced the same level of p-ERK and p-JNK1-2 in the fetus brain than LPS injection in the amniotic fluid alone. This point is not discussed by the authors (Figure 4B). In parallel, ischemic-reperfusion cycles induced the phosphorylation of IKK. Finally, pro-inflammatory cytokine expressions by RT-PCR were performed in the fetus brain and Interleukin-6 was measured in the pregnant mouse blood at different time-points. The vaginal exposure to LPS promoted pro-inflammatory response in the fetus brain in comparison with fetus only exposed to LPS in amniotic fluid. In parallel, vaginal LPS induced an earlier increase of IL-6 in the maternal blood than in the group that received only LPS in amniotic fluid.

Comments:

Introduction

The link between fetal brain and parkinson disease should be explained.

Experimental model and schedule

-There are lots of experimental conditions that reduced the clear understanding of the manuscript. The ischemic-reperfusion insult was finally less explored than the vaginal LPS challenge. It would be suitable to gain clarity to remove the ischemic-reperfusion experimental group.

- LPS in the amniotic fluid induced a high and rapid mortality that represented a strong challenge that did not reflect human condition.

Results

The result section did not give enough support to the reader to understand better the figures and the story reported in this manuscript.

Figures

All the figures are complex and demand lots of attention to the reader to interpret results. This is a really bad point of this manuscript.

Immunostainings are not convincing  particularly in Figure 3C and Figure 5B.

Figure 3A: mean weight of mouse fetuses was 850 +/- 150 grams ???

Figure 3C: which staining used to show apoptotic cells? No quantitative measure? 

In Figure 5E, the authors used a correlation coefficient. However, the X-axis corresponded to a qualitative (experimental conditions) and not quantitative variable. Therefore, from a statistical point of view, this analysis is not correct.

Discussion

This section is relatively short taking into account the huge amount of results.

Authors failed to clearly summarise their results and to give a place to their work in the current knowledge about the LPS action on immature brain and in the chorioamnionitis pathophysiology in humans.

Conclusion

The sentence that gave the final message of this manuscript did not seem in accordance with the rest of the manuscript and the primary aim supported by the authors in the introduction.

Author Response

Response to Reviewer 1 Comments

Point 1: Introduction

The link between fetal brain and parkinson disease should be explained.

Response 1: According to your request, we have revised the introduction.

Point 2: There are lots of experimental conditions that reduced the clear understanding of the manuscript. The ischemic-reperfusion insult was finally less explored than the vaginal LPS challenge. It would be suitable to gain clarity to remove the ischemic-reperfusion experimental group.

Response 2: Thank you for your comments. Because of our previous publication about ishemic reperfusion, there was less reported in this manuscript. However, this study used a continuous mouse model to comparably investigate the changes of molecular expression level in a classic pathology process of CAM. So, ischemic reperfusion may be a necessary challenge during pregnancy.

Point 3: LPS in the amniotic fluid induced a high and rapid mortality that represented a strong challenge that did not reflect human condition.

Response 3: Yes, it is the limitation of this study. Our experimental amniotic LPS induced a strong challenge maybe a systemic inflammation, although only LPS may not reflect the human condition completely.

Point 4: Results

The result section did not give enough support to the reader to understand better the figures and the story reported in this manuscript.

Response 4: According to your request, we have revised the results.

Point 5: Figures

All the figures are complex and demand lots of attention to the reader to interpret results. This is a really bad point of this manuscript.

Immunostainings are not convincing particularly in Figure 3C and Figure 5B.

Response 5: Thank you for your comments. We have revised the text for Figure 5B.

Point 6: mean weight of mouse fetuses was 850 +/- 150 grams ???

Response 6: Yes, they are selected fetuses.

Point 7: Figure 3C: which staining used to show apoptotic cells? No quantitative measure?

Response 7:  The staining method is the terminal deoxynucleotidyl transferase dUTP nick-end labeling assay, and apoptotic-like positive cells became brown color in Figure 3C.

Point 8: In Figure 5E, the authors used a correlation coefficient. However, the X-axis corresponded to a qualitative (experimental conditions) and not quantitative variable. Therefore, from a statistical point of view, this analysis is not correct.

Response 8: According to your comments, Figure 5E has been removed.

Point 9: Discussion

This section is relatively short taking into account the huge amount of results.

Response 9: According to your request, We have added a revised discussion.

Reviewer 2 Report

Dear Authors,

Thank you for the opportunity to review the manuscript „Amniotic LPS-Induced Apoptosis in the Fetal Brain Is Suppressed by Vaginal LPS Preconditioning but Is Promoted by Continuous Ischemic Reperfusion“.

The pathogenesis of cerebral pulsy is very important. The idea of the study is original.

The abstract adequately summarizes the manuscript. The key words are appropriate.

The introduction should be revised.

Line 30. Bacterial vaginosis (BV) has been experienced in more than 20% of reproductive-age 30

women in the world [1–4]. I suggest to use more recent literature (General population prevalence of BV is high globally, ranging from 23% to 29% across regions (Europe and Central Asia, 23%; East Asia and Pacific, 24%; Latin America and Caribbean, 24%; Middle East and North Africa, 25%; sub-Saharan Africa, 25%; North America, 27%; South Asia, 29%). (High Global Burden and Costs of Bacterial Vaginosis: A Systematic Review and Meta-Analysis. Peebles K, Velloza J, Balkus JE, McClelland RS, Barnabas RV. Sex Transm Dis. 2019;46(5):304.)

Line 33-35. Same as cerebral palsy, other disorders are caused by fetal brain dam- 33

age, which results from different risk factors including gestational age of <32 weeks, low 34

birth weight, asphyxia, infection, and inflammation induced by lipopolysaccharide (LPS). It is not clear – difference between infection and inflammation, I suggest to explain this (In case of infection (viral or bacterial) there are cytokines in response to bacterial or viral products. These products include lipopolysaccharide and others).

Line 39.  our mouse model was designed to mimic ascending inflammation at term. I suggest sto explain why You choose the term, but not preterm model, the cause of cerebral pulsy in term and preterm babies are different.

The results are present in the text and figures, are appropriately described.

The discussion. I suggest to add the strengths and limitations of this study.

Conclusions. I recommend to add more information.

Line 61-63 (introduction). The goal is to understand whether and how vaginal LPS regulates the vulnerability of the fetus to brain damage secondary to those risk factors that might be present in CAM, e.g., amniotic infection and ischemic reperfusion. The information in the conclussions part should answer the previous questions.

Author Response

Response to Reviewer 2 Comments

Point 1: The introduction should be revised.

Line 30. Bacterial vaginosis (BV) has been experienced in more than 20% of reproductive-age 30 women in the world [1–4]. I suggest to use more recent literature (General population prevalence of BV is high globally, ranging from 23% to 29% across regions (Europe and Central Asia, 23%; East Asia and Pacific, 24%; Latin America and Caribbean, 24%; Middle East and North Africa, 25%; sub-Saharan Africa, 25%; North America, 27%; South Asia, 29%). (High Global Burden and Costs of Bacterial Vaginosis: A Systematic Review and Meta-Analysis. Peebles K, Velloza J, Balkus JE, McClelland RS, Barnabas RV. Sex Transm Dis. 2019;46(5):304.)

Response 1: Thank you for your comments. According to your request, we have revised the text.

Point 2: Line 33-35. Same as cerebral palsy, other disorders are caused by fetal brain dam- 33 age, which results from different risk factors including gestational age of <32 weeks, low 34 birth weight, asphyxia, infection, and inflammation induced by lipopolysaccharide (LPS). It is not clear – difference between infection and inflammation, I suggest to explain this (In case of infection (viral or bacterial) there are cytokines in response to bacterial or viral products. These products include lipopolysaccharide and others).

Response 2: Thank you very much. According to your request, we have revised the text.

Point 3: Line 39.  our mouse model was designed to mimic ascending inflammation at term. I suggest sto explain why You choose the term, but not preterm model, the cause of cerebral pulsy in term and preterm babies are different. The results are present in the text and figures, are appropriately described.

Response 3: Thank you for your comments. According to your request, we have revised the text.

Point 4: The discussion. I suggest to add the strengths and limitations of this study.

Response 4: Thank you very much. According to your request, we have revised discussion.

Point 5: Conclusions. I recommend to add more information.

Line 61-63 (introduction). The goal is to understand whether and how vaginal LPS regulates the vulnerability of the fetus to brain damage secondary to those risk factors that might be present in CAM, e.g., amniotic infection and ischemic reperfusion. The information in the conclussions part should answer the previous questions.

Response 5: Thank you very much. We have revised the conclusion.
